# Engineered M13-Derived Bacteriophages Capable of Gold Nanoparticle Synthesis and Nanogold Manipulations

**DOI:** 10.3390/ijms252011222

**Published:** 2024-10-18

**Authors:** Joanna Karczewska-Golec, Kamila Sadowska, Piotr Golec, Jakub Karczewski, Grzegorz Węgrzyn

**Affiliations:** 1Department of Molecular Virology, Institute of Microbiology, Faculty of Biology, University of Warsaw, Miecznikowa 1, 02-096 Warsaw, Poland; joanna.karczewskagolec@gmail.com (J.K.-G.); piotr.golec@uw.edu.pl (P.G.); 2Department of Molecular Biology, Faculty of Biology, University of Gdansk, Wita Stwosza 59, 80-308 Gdansk, Poland; 3Nalecz Institute of Biocybernetics and Biomedical Engineering, Polish Academy of Sciences, Ks. Trojdena 4, 02-109 Warsaw, Poland; ksadowska@ibib.waw.pl; 4Advanced Materials Centre, Faculty of Applied Physics and Mathematics, Gdansk University of Technology, G. Narutowicza 11/12, 80-233 Gdansk, Poland; jakub.karczewski@pg.edu.pl

**Keywords:** gold nanoparticles, gold nanostructures, gold-binding peptide, phage display, gold precipitation, biogenic synthesis, bioreduction

## Abstract

For years, gold nanoparticles (AuNPs) have been widely used in medicine and industry. Although various experimental procedures have been reported for their preparation and manipulation, none of them is optimal for all purposes. In this work, we engineered the N-terminus of the pIII minor coat protein of bacteriophage (phage) M13 to expose a novel HLYLNTASTHLG peptide that effectively and specifically binds gold. In addition to binding gold, this engineered phage could synthesize spherical AuNPs of 20 nm and other sizes depending on the reaction conditions, aggregate them, and precipitate gold from a colloid, as revealed by transmission electron microscopy (TEM), atomic force microscopy (AFM), and scanning electron microscopy (SEM), as well as ultraviolet–visible (UV–vis) and Fourier-transform infrared (FTIR) spectroscopic methods. We demonstrated that the engineered phage exposing a foreign peptide selected from a phage-displayed library may serve as a sustainable molecular factory for both the synthesis of the peptide and the subsequent overnight preparation of AuNPs from gold ions at room temperature and neutral pH in the absence of strong reducing agents, such as commonly used NaBH_4_. Taken together, the results suggest the potential applicability of the engineered phage and the new, in vitro-identified gold-binding peptide in diverse biomimetic manipulations.

## 1. Introduction

With their wide range of applications in medicine and industry, gold nanoparticles (AuNPs) have been among the most intensively explored structures in recent years. Their intriguing, tunable physicochemical properties are useful for electronics, sensor technology, catalysis, as well as diagnostics and therapeutics (including probes, therapeutic agents or therapeutic agent carriers, photothermal and contrast agents) [1,2,3,4]. The specific size and shape of AuNPs determine their activity in these applications [5,6].

In terms of biomedical applications, gold nanoparticles—of all nanomaterials—are the most intensively researched ones [7] and, like AuNPs in industrial uses, have already moved from laboratory research to real-life applications. One example is gold nanoparticle-based SARS-CoV-2 rapid antigen tests, which have been approved for clinical use and widely adopted by the market [8]. The other example is over 20 ongoing and completed clinical trials worldwide that have implemented gold NPs either as a drug delivery system (in glioblastoma; breast cancer; type I diabetes; unspecified solid tumors), for photothermal therapy (in atherosclerosis; head and neck, prostate, or lung cancer; acne; dry eye syndrome), or as a therapeutic agent (in multiple sclerosis; amyotrophic lateral sclerosis; and Parkinson’s disease), among other diseases [9]. Two AuNP-based nanoformulations have already been approved by the Food and Drug Administration (FDA) agency, i.e., AuroLase (Nanospectra Biosciences), PEG-coated silica-gold nanoshells for thermal ablation of solid primary and/or metastatic lung tumors, and NU-0129 (Northwestern), nucleic acids arranged on the surface of a spherical gold nanoparticle for the treatment of glioblastoma [7].

Several methods were proposed for the preparation of nanoparticles. One common chemical method for synthesizing AuNPs is the reduction of gold salts, first described in 1951 [10]. The technique was subsequently improved to increase the yield and to control the size of AuNPs [11,12,13,14]. Other methods were also described, including physical reduction, photochemical reduction, biological reduction, solvent evaporation techniques, and the microwave irradiation approach. Detailed descriptions of the strategies for the synthesis of gold nanoparticles are provided in the recently published reviews [1,15,16,17,18]. Chemical, physical, and physicochemical methods for AuNP synthesis require the use of hazardous chemicals, thermal energy, or specialized equipment. The resulting set of their limitations includes biocompatibility and environmental and economic considerations.

There is an ongoing need in both industrial and scientific settings to develop nontoxic, simple, and robust methods for the synthesis of AuNPs of various sizes and shapes. Biosynthesis and assembly of nanoparticles with the use of biomolecules and microorganisms may be an answer and have emerged as viable alternatives to the methods mentioned above [19,20]. Such biomolecule-based approaches have been increasingly used due to stringent environmental regulations and to maintain the biocompatibility of AuNPs for downstream applications [17]. However, although these methods employ biomolecules as one of the reagents in AuNP synthesis, in most cases they also require large amounts of hazardous, highly toxic chemicals; strong, conventional reducing agents such as NaBH_4_; or harsh conditions, including extreme pH [21]. For example, peptides that have been commonly used as green alternatives for AuNP preparation have in fact been routinely obtained by chemical synthesis methods such as Boc and Fmoc solid-phase syntheses, which involve the use of large volumes of hazardous solvents, rendering the “greenness” of the subsequent peptide-directed synthesis of AuNPs questionable [22,23]. Therefore, there is still room for improvement in such biogenic methods of AuNP synthesis.

Filamentous viruses are also among such potential biomolecules for nanoparticle synthesis, and they have already proved applicable in creating nanoparticle arrays or in controlling the size and shape of the synthesized structures [24,25]. Benefits associated with the use of viruses, such as their simple replicative growth, liquid crystalline-like self-assembly, and the simplicity of genetic modifications, lay the groundwork for the development of (multi)functionalized virus derivatives [26,27,28].

The M13 bacteriophage (phage), a filamentous virus that infects *Escherichia coli* bacteria carrying the F-episome, belongs to the *Inoviridae* family and has a capsid of ~880 nm in length and ~6 nm in width [29]. From the perspective of phage biologists, the most distinctive M13 feature is perhaps its infection strategy, as this chronic phage undergoes neither a lytic nor a temperate life cycle [30]. In turn, nanoscience and nanoengineering researchers look upon M13 as an interesting, naturally occurring, monodisperse, self-assembling nanofiber [31].

The major virus coat protein, pVIII, is arranged in a right-handed fish-scale pattern of approximately 2700 copies surrounding single-stranded DNA (ssDNA). Another coat protein—pIII—in three to five copies—forms a distal part of the capsid. Both pIII and pVIII proteins have been commonly used to expose foreign peptides on a phage surface in a well-established, combinatorial phage display technique [32]. Traditionally, in this approach, nucleotide sequences encoding foreign peptides were inserted into the sequences of phage *pIII* and/or *pVIII* gene(s). The resulting recombinant phages were propagated to generate peptide libraries. In the next step of phage display, screening of such libraries, which present a huge variety (~10^9^) of random peptide sequences, allows for a high-throughput selection of peptides with the desired binding properties, without a priori knowledge. The desired phage-displayed peptides are separated by binding affinity purification and their sequence is obtained by DNA sequencing of the corresponding phage clones [31,33,34]. Originally developed to display peptides, the phage display method has evolved over the years to present antibodies [35], T-cell receptors, enzymes, proteomes, secretomes, and the so-called alternative affinity scaffolds [36]. Phage display has already proved useful in the selection of functional biomolecules for some of the most challenging targets in the biomedical field. Several phage display-derived peptides were approved by the FDA for the treatment of various diseases [37].

In this work, with the use of phage display, we aimed to develop an engineered M13 phage that exposes a new, in vitro-identified Au-binding peptide capable of synthesizing AuNPs overnight under benign conditions. Such phages could be employed for biogenic, bottom-up synthesis and manipulation of gold nanostructures particularly for further biological applications, which have considerable concerns over contamination.

## 2. Results

### 2.1. Isolation of Phages Exposing Au-Binding Peptides

The isolation of phages that expose Au-binding peptides was carried out with the use of a commercially available Ph.D.™-12 Phage Display Peptide Library (NEB). Three rounds of a biopanning procedure were performed, and 20 phages were randomly chosen for sequencing. Twelve of the phage clones contained a foreign peptide insertion within the pIII protein. The identified peptide sequences are presented in Table 1. Sequence analysis revealed the same peptide sequence (HLYLNTASTHLG) in phages M13 Au-6 and M13 Au-9. One such phage clone (M13 Au-6) was therefore subjected to further analyses.

### 2.2. Binding Efficiency of the Selected Peptide to a Gold Substrate

To determine the Au-binding efficiency of the selected phage clone, an output-to-input ratio (O/I) analysis was performed, as described in Section 4.3. For the analysis, the M13 Au-6 phage clone exposing the HLYLNTASTHLG peptide was chosen and the wild-type M13KE phage was used as a control. The result showing a very high Au-binding efficiency by the Au-6 phage clone versus the wild-type phage without a foreign peptide is presented in Figure 1. To test the binding specificity of the M13 Au-6 phage clone exposing the HLYLNTASTHLG peptide, we also analyzed ZnO nanoparticles as a substrate (binding partner) (Figure 1). In addition, by comparing the output/input (O/I) ratios, we found that the phage-presented dodecapeptide identified in this study had a higher affinity for gold than a previously reported, phage-display-selected TLLVIRGLPGAC dodecapeptide, which had been characterized as having a strong gold-binding affinity [38]. With an O/I ratio of approximately 7 × 10^−3^, the phage-presented HLYLNTASTHLG peptide selected in our study exhibited approximately seven times higher gold-binding efficiency than its TLLVIRGLPGAC counterpart, which had an O/I ratio of approximately 1 × 10^−3^.

In the next step, the affinity of the M13 Au-6 phage for gold was investigated by mixing a preformed gold colloid with the phage suspension, as described in Section 4.5. After the addition of the gold colloid to the phage suspension, aggregation of the nanoparticles occurred, which was directly observed as a color change from deep red to bluish (see Figure 2, inset), and confirmed by absorption spectroscopy measurements (Figure 2). Gold nanoparticles exhibit a distinctive optical feature commonly referred to as localized surface plasmon resonance (LSPR), which is a result of the collective oscillation of conduction electrons on the metal surface when excited by light. In the case of spherical AuNPs, their optical properties are highly dependent on both the diameter of the nanoparticles themselves and the physical properties of their surroundings. Smaller gold nanospheres primarily absorb light, with absorption bands around 520 nm (as can be seen from Figure 2, curve a). When AuNPs form agglomerates of higher dimensions than individual nanoparticles, they exhibit enhanced scattering, with bands broadening and shifting towards longer wavelengths (a phenomenon known as red-shifting) (see Figure 2, curve b). This enhanced scattering in agglomerated particles is due to both their larger optical cross-sections and the increase in the ratio of scattering to total extinction with increasing particle size. As mentioned above, the optical properties of gold nanoparticles are also affected by the refractive index of their surrounding environment. When the refractive index near the nanoparticle surface increases, the absorption band of the nanoparticle also shifts towards longer wavelengths (a red-shift). The red-shift of c.a. 30 nm, observed in the UV–vis spectrum of a mixture of AuNPs and phages relative to the spectrum of AuNPs alone, can be attributed to the enhanced scattering observed for agglomerates and to the increase in the refractive index near the AuNP surface resulting from the interaction between the phages and gold. Such an observation reflects the effect of nanoparticle size and local refractive index on the optical properties of AuNPs which is widely applied in the colorimetric detection of biological molecular interactions that alter the aggregation and dispersion status of particles [39]. For comparison, the UV–vis spectrum of the phage suspension is also shown in Figure 2 (curve c), revealing no distinct absorption band in the analyzed region.

SEM images of Au nanoparticles, before and after the addition to phages exposing Au-binding peptides, are shown in Figure 3. Note that both SEM images were captured at the same magnification to clearly show the size differences of AuNPs before and after the addition of the preformed gold colloid to the engineered phage suspension. Nanogold particles are nearly spherical, with diameters of approximately 20–25 nm, and are well dispersed (Figure 3A). The binding of AuNPs by the M13 Au-6 phage clone, resulting in the aggregation of Au nanoparticles, is clearly visible (Figure 3B). The aggregates formed consist of several individual nanoparticles, and the aggregate dimensions are in the range of 20–100 nm. Note that the blurring in the images is due to the presence of phages/peptides surrounding the AuNPs. Due to the nature of the material, which is mostly organic, and because we wanted to avoid destroying the phage–Au bioconjugates, no special manipulation (which is common for other, less fragile materials) was performed prior to SEM imaging; this resulted in the “clouding” effect, but did not affect the interpretation of the results.

Interestingly, when the procedure was reversed, i.e., the phage suspension was added to the preformed gold colloid, very rapid and effective precipitation of gold nanoparticles was observed. The supernatant became clear and a brown pellet was formed at the bottom of the flask. Phages and nanoparticles formed bundled aggregates that were too large to form a suspension in water, thus the precipitation occurred.

Further studies of the genetically engineered phages for their Au-binding ability were carried out using atomic force microscopy (AFM) (Figure 4) and transmission electron microscopy (TEM) (Figure 5). Figure 4 presents the interactions between a glass surface covered by gold colloid (see Section 4.4) and the M13 Au-6 phage clones. In Figure 4A, the structure formed by Au nanoparticles on the glass support is presented. In Figure 4B–D, phages exposing Au-binding peptides with Au nanoparticles on the glass support are shown. Agglomerates of the phage virions around the gold nanoparticles can be observed. Separated gold nanoparticles of 10–20 nm were surrounded by larger agglomerates of 50–200 nm (Figure 4B). Looking at the system at a lower magnification, as shown in Figure 4C,D, longitudinal structures formed by clusters of phages with uniformly dispersed, individual gold nanoparticles can be clearly seen.

Figure 5 presents a TEM image of interactions between M13 Au-6 phage clones (long filamentous virions of M13 phages) and AuNPs (black spots). Many AuNPs colocalized with the terminal parts (ends) of the filamentous phage virion structures. This observation confirmed that it is likely that M13 Au-6 phage clones interact with AuNPs through a virion tip where the modified pIII protein is located.

### 2.3. Phage-Assisted Synthesis of AuNPs

After demonstrating that the engineered phages obtained have remarkable binding affinity for the Au surface, their inherent ability to synthesize Au nanoparticles from the HAuCl_4_ precursor was explored (see Section 4.6). Au nanoparticles were obtained overnight from the HAuCl_4_ precursor in the reaction with engineered phages (M13 Au-6, 10^11^ pfu/mL) at room temperature (RT) both in the presence of ammonia and triethylamine (TEA) which were used to adjust pH to neutral. These reaction mixtures are shown in the inset of Figure 6—the vials containing AuNPs synthesized in the presence of ammonia and TEA are presented on the left and right, respectively. On the other hand, the reduction of Au(III) from the HAuCl_4_ precursor to bulk metallic gold was observed in the case of both blank (control) experiments that were carried out without the phages exposing the Au-binding peptides. These reactions contained the HAuCl_4_ precursor and ammonia or TEA. The reaction mixtures were transparent with dark sediments at the bottom of the flasks. SEM analysis of the pellets revealed that no nanoparticles were formed in these control experiments.

UV–vis spectroscopic analysis (Figure 6) of the samples synthesized in the presence of ammonia revealed an absorption band at 530 nm and a side arm at 600 nm, indicating that some of the nanoparticles of ca. 30 nm are well dispersed in the solvent, while others form agglomerates. This is in agreement with the color of the sample, which is purple. In the second sample, where AuNPs were synthesized in the presence of TEA, the UV–vis spectrum shows an absorption band peaked at 580 nm, which corresponds to larger nanoparticles. The color of the sol sample is orange with violet opalescence.

Fourier-transform infrared (FTIR) spectra were recorded for dried Au nanoparticles and the Au-binding phages. The results are presented in Figure 7. The spectrum recorded for M13 Au-6 phages (curve c) reveals characteristic bands of peptides, i.e., NH stretching (3300 cm^−1^), first amide band at 1650 and 1625 cm^−1^, second amide band at 1540 and 1525 cm^−1^, and third amide band at 1280 and 1240 cm^−1^. The bands referring to OH bonds at ~1400 and the split band between 1140–1000 cm^−1^ are clearly visible, as the peptide sequence is rich in hydroxy amino acids. The band at 2880 cm^−1^ may be assigned to CH bonds in the amino acid backbone or to -NH_3_^+^, which is plausible as it disappeared after the AuNP synthesis. In the case of spectra of M13 Au-6 phages with gold nanoparticles (curves a and b), all bands referring to OH bonds are shifted to lower wavenumbers and the bandwidth is narrower. Similarly, the I, II, and III amide bands are narrower without splitting and are shifted by about 10–20 cm^−1^.

One can assume that hydrogen bonds formed within the amide and hydroxyl groups in phage-exposed, gold-binding peptides are destroyed after the nanoparticle synthesis. This could indicate that hydroxyl groups are involved in the gold surface binding, which is consistent with the results presented by other groups [40,41]. The spectra a and b in Figure 7 are identical, hence the influence of ammonia or TEA on the phages is negligible, if any.

SEM imaging was used to determine the size and shape of the nanoparticles obtained. In both cases, i.e., when either ammonia or TEA was used, the nanoparticles were nearly spherical and formed agglomerates. However, their sizes were substantially different. Nanoparticles synthesized in the ammonia environment were smaller compared to those obtained in the presence of TEA, which is in agreement with the results obtained by UV–vis spectroscopy (Figure 6). In Figure 8, SEM images of Au nanoparticles synthesized by engineered phages in the presence of ammonia are presented. The average size of AuNPs was estimated to be 20 nm.

Figure 9 shows SEM images of Au nanoparticles synthesized by engineered phages in the presence of triethylamine. In this case, the nanoparticles formed were larger than in the experiments with ammonia, with a size range of 80–120 nm.

Figure 10 presents TEM imaging of AuNPs synthesized by the M13 Au-6 phage. The synthesized AuNP (black spot) is located at the end of the phage virion structure where the modified pIII protein is located. In the control experiment with a wild-type M13KE phage, no AuNPs were observed.

As mentioned above, in the images showing the interaction between the M13 Au-6 phage clone and AuNPs (Figure 5 and Figure 10), the location of AuNPs on the phage virion tip was observed. However, based on the TEM images it is difficult to unambiguously confirm that direct interactions between the pIII protein and AuNPs occur. Therefore, the synthesis of AuNPs using the Au-binding peptide only was performed. Figure 11 shows the SEM image of Au nanoparticles synthesized from a precursor by the HLYLNTASTHLG peptide only, in the presence of triethylamine. In this case, the nanoparticles obtained formed agglomerates and were in the size range of 80–120 nm. This provides indirect evidence that the phage-display-selected Au-binding peptide (HLYLNTASTHLG) exposed on the pIII protein is required for phage-directed AuNP synthesis. In a control experiment with an Eu_2_O_3_-binding peptide, SRTGNWTRIDQS [24], no AuNPs were observed.

## 3. Discussion

The dual plasmonic and catalytic functionalities, combined with the ease of tailoring their surface chemistry and the presumed lowest toxicity among metal nanoparticles [42], make AuNPs valuable in diverse industrial and biomedical fields. Although various experimental procedures have been reported for the AuNP preparation, none of them is optimal for all purposes. Therefore, effective new methods for AuNP synthesis are desirable. Among these, biological systems seem to offer interesting possibilities. In this report, we used phages exposing peptides selected by a combinatorial phage-display approach which we had successfully used previously [24,25,43], as sustainable molecular factories to produce AuNPs from a gold ion precursor in the absence of strong, chemical reductants and under benign conditions. To the best of our knowledge, this is the first report of AuNPs obtained overnight at neutral pH and room temperature using an engineered M13 phage system that simultaneously (i) relies on a novel Au-binding peptide experimentally selected in vitro for its high gold-binding affinity (as opposed to using an arbitrarily chosen, potentially binding amino acid sequence [44,45]); (ii) utilizes the properties of the engineered phage itself (as opposed to the use of a strong chemical reductant) in the preparation of AuNPs from HAuCl_4_; and (iii) enables biogenic synthesis of the Au-binding peptide by propagating the engineered phage in a bacterial culture, which is an easily scalable and environmentally benign method.

Many research efforts on gold-binding peptides, including those exposed on phages, and on their role in gold nanoarchitecture assembly and/or nanoparticle formation were reported previously. An early study described gold recognition by repeating polypeptides containing several direct repeats of identical 14-mer or 28-mer peptides displayed on *E. coli* cell surfaces as part of the LamB porin [40]. Although it had been reported that cysteine pairs forming disulfide bridges have a high affinity for the gold surface [46], the LamB-exposed, gold-binding peptides were not cysteine-rich while being rich in hydroxy amino acids—serine and threonine [40]. In fact, the gold-binding peptide identified in our study is devoid of cysteine residues, whereas it contains hydroxy amino acids (Table 1). In another work, highly engineered, heterofunctional phage templates were created for material assembly, including the arrangement of preformed AuNPs into 1D arrays [41]. However, contrary to the results presented in this report, neither the engineered phage alone nor the selected gold-binding peptide alone could synthesize nanoparticles. The identified Au-binding peptide sequence contained four serine residues, each with a hydroxyl group on the side chain [41]. Thus, just as in the previous report [40], as well as in our study (Table 1, Figure 7), the hydroxyl-rich peptides showed a high affinity for gold lattices [41]. Subsequently, a dodecapeptide, selected from an M13 phage-displayed peptide library, was used for gold nanoparticle synthesis under ambient reaction conditions [47]. This gold-binding dodecapetide, with a calculated pI of 5.18, contained a central hydrophobic tetrapeptide (VLIA) flanked by two polar tetrapeptides (TGTS and TPYV) [47]. The peptide did not contain sulfur- or amine-containing amino acid residues known to covalently bind to the surface of gold particles [48], or other motif sequences previously found in gold-binding proteins [40]. While in our report the engineered phage was utilized for biogenic synthesis of the gold-binding and -forming peptide, in the previous study [47] an additional experimental step after the phage-display peptide selection—chemical synthesis of the peptide—was required to complete the peptide-directed synthesis of gold nanostructures.

While the above-mentioned studies and our work identified the new gold-binding peptides in vitro, other studies used a peptide whose amino acid sequence was arbitrarily chosen for AuNP synthesis [44,45]. For example, five tyrosine residues were chosen for peptide-directed AuNP synthesis [45] as the critical role of this amino acid in peptide-assisted AuNP synthesis was demonstrated elsewhere, both in vitro (where even a single tyrosine residue reduced gold ions) [49] and by molecular dynamics simulations [50]. The HLYLNTASTHLG peptide identified in our work contains a tyrosine residue.

Recent efforts to assemble functional gold nanosystems with the use of engineered phages include the creation of AuNP–M13 hybrid complexes that retain their natural recognition/targeting ability and whose tropism toward *E. coli* can be tailored based on the nanoparticle size [51] or the engineering of the M13 phage to both display the receptor-binding proteins of various phages and to bind AuNPs for surface plasmon resonance-based detection of human and plant pathogenic bacteria [52].

One fresh approach to accelerating gold-binding peptide discovery for biomimetic nanostructure fabrication is the integration of machine learning (ML) models into peptide design. This can be exemplified by recent reports [53,54], where the experimental binding data of 1720 and 860 decapeptides toward gold nanoparticles [55] were used for training and testing two ML models to predict the gold-binding ability of new peptides, classifying them as either strong or weak binders of AuNPs. Interestingly, peptide hydrophobicity was found to have the greatest influence on the classification of a decapeptide as a strong gold binder. As with any model, the patterns discovered through data mining [53,54] have inherent limitations, one of which is the applicability of the rules, especially the sequence rules, only to decapeptides. With this in mind, it is worth noting that the HLYLNTASTHLG gold-binding dodecapeptide identified in our study is indeed hydrophobic. In silico calculations classify it as moderately hydrophobic and place it in the very middle of the peptide hydrophobicity classification.

Both the sequence and the calculated pI of the peptide identified in our work (Table 1) were different from those of previously reported gold-binding peptides selected through phage or cell surface display [41,47,56]. These results indicate that HLYLNTASTHLG represents a distinct class of AuNP-binding and -forming peptides. Interestingly, while this sequence was not reported previously as a gold-binding one, it did appear in a study on phage-display-selected shuttle peptides crossing the blood–brain barrier (BBB) in an in vitro model [57]. As mentioned in the Introduction (Section 1), gold nanostructures have been increasingly investigated as therapeutic agents, drug delivery systems, or photothermal agents in neurodegenerative disorders and brain cancers. Hence, the peptide described in this report (HLYLNTASTHLG) might be considered a potential carrier of drugs to be delivered to the brain.

The peptide exposed on the surface of phages described in this report is capable of both binding Au and synthesizing AuNPs in the absence of strong reducing agents, such as NaBH_4_. While Au-binding peptides were experimentally identified previously, the sequence of the peptide identified in this work differs from those reported previously [45,47,56,58]. This confirms the previous suggestion that there are many possible variants of oligopeptides that can bind the same metal or metal derivative [33]. Importantly, we demonstrated the selective binding of the HLYLNTASTHLG peptide to gold nanoparticles rather than to other nanomaterials (Figure 1 and Figure 11). This binding was about seven times more efficient than that previously described for another peptide (reported as having a strong gold-binding affinity [38]). Thus, it appears that the method and the obtained recombinant phage exposing the HLYLNTASTHLG peptide on the virion surface could be useful in all processes and technologies requiring gold-binding, as they are more efficient in this aspect than previously reported peptides.

Engineered phages (10^11^ pfu/mL) formed AuNPs from the gold ion precursor overnight at RT and neutral pH in the presence of ammonia or TEA (both used for pH adjustment) and in the absence of strong reducing agents, such as NaBH_4_. A control experiment with the HAuCL_4_ precursor and ammonia or TEA but without the engineered phages (M13 Au-6) resulted in the reduction of Au(III) from the precursor to bulk metallic gold. The reaction mixtures were transparent with dark sediments at the bottom of the flasks. SEM analysis of the pellets also confirmed that no nanoparticles were formed, indicating that the engineered phage presenting the HLYLNTASTHLG peptide was necessary for the synthesis of AuNPs from HAuCl_4_. It was shown previously that amino acids can act as both reducing and capping agents in the formation of AuNPs [59,60] and references therein. Furthermore, all amino acids except cysteine have the ability to reduce HAuCl_4_ to gold NPs [61,62]. Importantly, in the control experiment with a wild-type M13KE phage and gold ion precursor, no AuNPs were formed.

The gold-binding phages constructed in this work, despite some similarities to those reported previously, have also very specific properties. Upon addition of a preformed gold colloid to a suspension of phages exposing the Au-binding peptide (HLYLNTASTHLG), aggregation of the nanoparticles occurred. Intriguingly, when the procedure was reversed, i.e., when the phage suspension was added to the gold colloid, a very rapid and efficient precipitation of gold nanoparticles was observed. The supernatant became clear and a pellet was formed at the bottom of the flask. One could suggest that such a phenomenon may be used to recover nanoparticles from sludge and purify water from gold nanoparticles. According to the previously published data, there are currently over 1800 consumer products containing engineered nanoparticles [63]. Nanoparticles can be released into the environment during the manufacturing processes, as well as during the utilization and degradation of the spent products. Thus, the phage-assisted precipitation of nanoparticles may be useful in wastewater treatment. Such a strategy has already been shown to be effective for gold ion reduction and gold retrieval [64].

Interestingly, some of the biomedical and environmental applications of gold nanostructures require, and capitalize on, aggregated AuNPs. Aggregates of noble metal nanoparticles exhibit enhancement for many detection and treatment techniques compared to individual nanoparticles. Aggregation-induced emission enhancement (AIEE) is a phenomenon in which luminophores such as noble metallic nanoclusters exhibit higher photoluminescence efficiency in the aggregated state than as a homogeneous dispersion. In a recent report, the AIEE of gold nanoclusters (AuNCs) prepared by peptide-directed, NaBH_4_-mediated reduction of HAuCl_4_ was investigated, and the AuNCs were successfully employed as luminescent probes for imaging intracellular lysosomes by leveraging their pH-responsive AIEE behavior [65]. In addition, the aggregation of metal nanoparticles plays an important role also in surface-enhanced Raman spectroscopy (SERS), and it was shown that the formation of gold nanoaggregates provided optimal SERS enhancement for ultrasensitive probing inside the endosomal compartment [66]. Another example of the use of AuNP aggregation comes from gold nanoparticles used as effective photothermal agents (PTAs) in cancer therapy. Targeted aggregation of gold nanoparticles prolongs tumor accumulation and retention of PTAs and improves photothermal conversion efficiency, resulting in increased efficiency of tumor photothermal therapy [67].

In conclusion, we constructed an M13-derived bacteriophage that exposed a foreign peptide capable of specifically binding gold. This phage–peptide system combined the ability of the pIII phage coat protein to self-assemble into a well-defined architecture with the precise gold recognition feature of the HLYLNTASTHLG peptide. The phage–peptide system could synthesize AuNPs (in the absence of strong reductants, such as NaBH_4_), aggregate them, as well as precipitate gold from a colloid and appears to be potentially useful for various manipulations of gold nanostructures. Unlike previous reports on the fabrication of gold nanoparticles using phage-display-selected peptides, this study features a gold-binding phage that has a dual role in the syntheses. First, propagation of the peptide-presenting phage enables biogenic synthesis of the desired peptide. Second, the propagated, peptide-presenting phage then enables the fabrication and manipulation of gold nanoparticles. In contrast, previous works on the fabrication of gold nanostructures reported that the selected peptide was chemically synthesized after the phage display selection step, or the peptide-presenting phage was used only as a scaffold for the assembly of preformed AuNPs into other nanostructures as the synthesis of AuNPs from the HAuCl_4_ precursor required a strong, chemical reductant.

## 4. Materials and Methods

### 4.1. Materials

All chemical reagents were of analytical grade and were purchased from Sigma-Aldrich (St. Louis, MO, USA), unless otherwise noted. The wild-type M13KE phage, the Ph.D.™-12 Phage Display Peptide Library, and *E. coli* ER2738, all from New England Biolabs (NEB, Ipswich, MA, USA), were used.

### 4.2. Isolation and Identification of the Au-Binding Peptide

Au-binding peptides were isolated from the Ph.D.™-12 Phage Display Peptide Library (NEB) in three rounds of a biopanning procedure. An Au-coated glass plate (further referred to as glass Au), prepared as described in Section 4.4, was used to bind phages exposing the peptides of interest. Glass Au was incubated with TBST buffer (50 mM Tris-HCl pH 7.5, 150 mM NaCl, 0.1% Tween-20) for 30 min. Next, 10 µL of phage peptide library (10^11^ phage virions exposing random peptides) was added to 90 µL of TBST buffer and then dropped onto the glass Au surface. After 30 min of incubation at room temperature (RT), the gold surface was carefully rinsed several times with TBST buffer. The bound phages were eluted with 1 mL of 0.2 M glycine-HCl pH 2.2. The eluted phages, after their propagation (carried out according to the NEB protocol), were used in the next biopanning procedure. After the third biopanning step, the eluted phages were titrated on double agar plates containing X-gal/IPTG. Phages from single plaques were propagated, and single-stranded DNA was isolated according to a previously reported procedure [68] and submitted for sequencing with the use of 96 gIII sequencing primer (NEB).

### 4.3. Analysis of Au-Binding Efficiency by the Phage-Exposed Peptides

A phage suspension in TBS at a concentration of approximately 10^8^ pfu/mL was dropped onto the glass Au (prepared as described in Section 4.4). This suspension was called the input number of phages (I). After 30 min of incubation at RT and several rinses with TBS, the phages were eluted with 0.2 M glycine pH 2.2 buffer. The concentrations of the eluted phages were calculated with the use of a titration procedure (NEB). The resulting number was called the output number of phages (O). The binding efficiency was calculated as an O/I ratio, as described previously [69].

### 4.4. A Chemical Method for Gold Colloid Synthesis

Gold colloid was prepared as described previously [70]. Briefly, 30 mg of HAuCl_4_·xH_2_O (~49% wt. Au, Cat#254169, LOT#MKCK6653) was dissolved in 150 mL of deionized water and heated under reflux for 5 min. Then, 1.8 mL of trisodium citrate solution (50 mg/mL) was added with vigorous stirring. Heating was continued until a stable purple color was obtained (c.a. 2 h). The gold colloid was kept at RT in the dark and used in further experiments. The gold colloid obtained was also used for the preparation of glass Au (Section 4.2). The gold colloid was dropped onto a clean glass slide (2.5 × 2.5 cm) and left to dry at RT.

### 4.5. Binding of Phages to a Golden Substrate

In order to analyze the affinity of phages for gold surfaces, the interaction of phages with gold nanoparticles, being in the form of a colloid or deposited on a support, was studied. In the first approach, 1 mL of gold colloid was added to 20 mL of phage suspension (10^11^ pfu/mL), or the phage suspension was added to the gold colloid. The samples obtained were examined by scanning electron microscopy (SEM) and UV–vis spectroscopy. In the second approach, the previously obtained glass Au was used (see Section 4.4). The gold-coated plate was immersed in the suspension of phages (10^11^ pfu/mL) exposing Au-binding peptides, incubated for 10 min, rinsed several times with deionized water, and left to dry. These samples were characterized by atomic force microscopy (AFM).

### 4.6. Phage- and Peptide-Assisted Synthesis of AuNPs

For AuNP synthesis, we used both the phages presenting the selected peptide on the pIII protein and the selected peptide only. In the phage-assisted synthesis, the HAuCl_4_ solution was prepared (0.024 g of HAuCl_4_·xH_2_O, containing ~49% of Au (Cat#254169, LOT#MKCK6653) in 10 mL of deionized water) and ammonia or triethylamine was added dropwise to adjust the pH to neutral. Then, 20 mL of a suspension containing 10^11^ pfu/mL of phages was added to each solution and the mixtures were left overnight at RT. The HAuCl_4_ solution with ammonia or triethylamine, but without the phages exposing the Au-binding peptides or with a wild-type M13KE phage, was also left overnight (control experiments). For the synthesis of AuNPs with the use of the selected peptide only, the solution of HLYLNTASTHLG peptide (0.5 mg/mL) was added to the HAuCl_4_ solution, and the pH was adjusted to neutral with TEA. The mixture was kept overnight without stirring. To increase the monodispersity of the obtained AuNPs, ascorbic acid solution (2 mM) was added. In the control experiments, the same amounts of HAuCl_4_ and C_6_H_8_O_6_ solutions were mixed with a different peptide (SRTGNWTRIDQS), which allows the synthesis of Eu_2_O_3_ nanoparticles [24], or incubated without any peptide, in the presence of TEA.

### 4.7. Microscopic Analyses

For scanning electron microscopy (SEM) using the Quanta 250 FEG scanning electron microscope (FEI, Hillsboro, OR, USA), aqueous solutions of the samples were dropped onto the conductive carbon support and left to dry overnight at RT.

Atomic force microscopic (AFM) images were obtained with a MultiMode AFM instrument using a Nanoscope V controller (Bruker, Billerica, MA, USA) operating in the tapping mode with an uncoated RFESP tip (Bruker). The spring constant of the cantilever was 3 N/m. The scan area was 5 × 5 µm. Raw AFM data were processed with the Gwyddion 2.32 SPM data analysis and visualization tool.

Two samples were analyzed, namely, gold-covered glass slides, prepared as described in Section 4.4, before and after the immersion in the suspension of engineered phages (10^11^ pfu/mL). The immersed slide was incubated for 10 min, rinsed several times with deionized water, and left to dry.

For transmission electron microscopy (TEM) analysis, AuNPs synthesized by the M13-derived bacteriophages exposing the selected HLYLNTASTHLG peptide were placed on grids coated with a 2% collodion solution and carbon. AuNPs were negatively stained with 2% uranyl acetate or examined without staining using a Philips CM100 electron microscope (Philips/FEI Corporation, Eindhoven, The Netherlands) at 80 kV.

### 4.8. UV–Vis and FTIR Spectroscopy

UV–vis spectra were measured using Lambda 10, PerkinElmer (Waltham, MA, USA). Absorption spectra of aqueous solutions in quartz cuvettes were recorded in the wavelength range of 200–900 nm.

Fourier-transform infrared (FTIR) spectra were recorded at room temperature using a Perkin–Elmer spectrometer (model Frontier FTIR MIR/FIR, Waltham, MA, USA). The FTIR spectra of the dried samples were collected in the attenuated total reflectance (ATR) mode in the wavenumber range of 4000–400 cm^−1^.

## Figures and Tables

**Figure 1 ijms-25-11222-f001:**
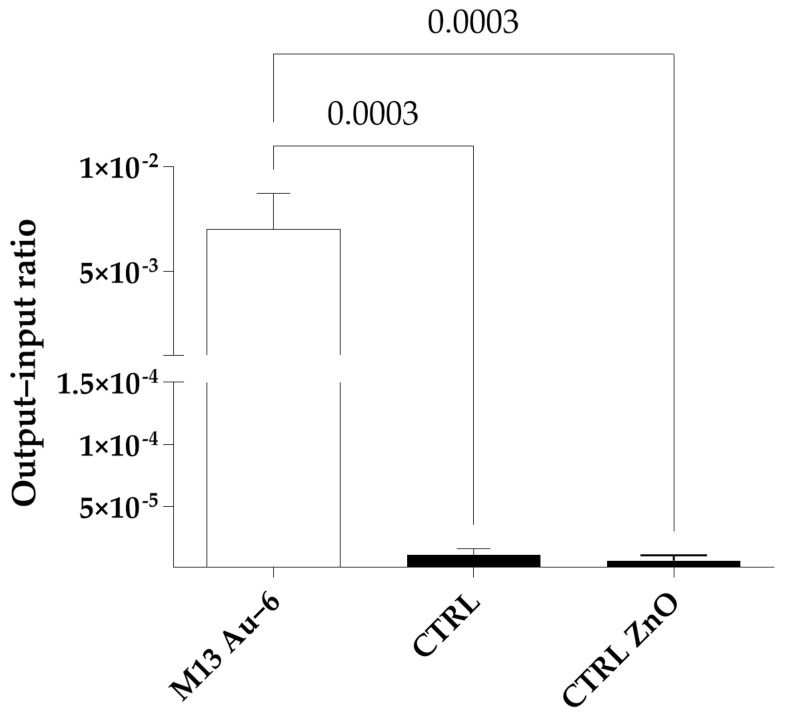
Binding efficiency (output/input (O/I) ratio) to the gold surface by the selected M13 Au-6 phage exposing the HLYLNTASTHLG peptide. Wild-type M13KE phage and the gold surface (CTRL) and M13 Au-6 and ZnO nanoparticles (CTRL ZnO) were used as controls. The results shown are average values of three experiments, with SD represented by the error bars. The difference between the values obtained for M13KE (control) and M13 Au-6 was statistically significant (*p* = 0.0020). One-way ANOVA was used for statistical analysis.

**Figure 2 ijms-25-11222-f002:**
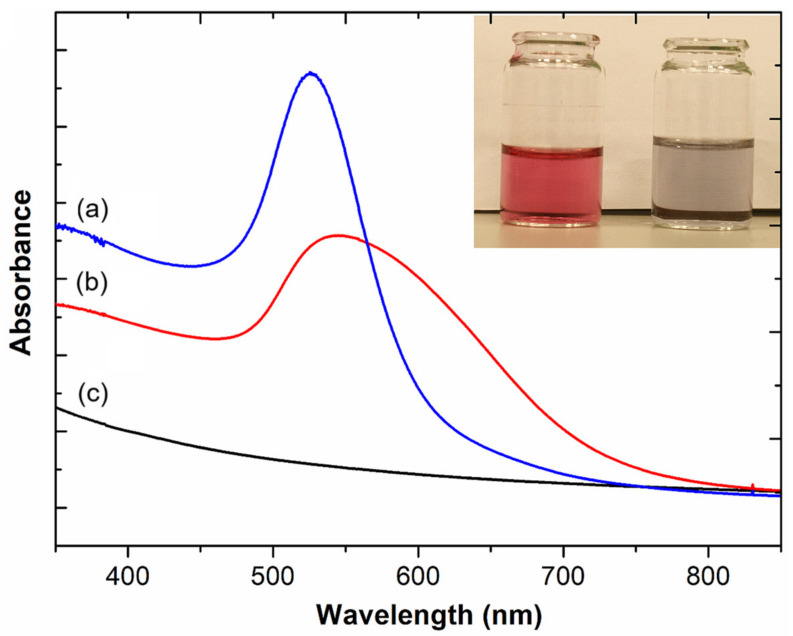
UV–vis spectra of (a) nanogold, (b) M13 Au-6 with nanogold, and (c) M13 Au-6 suspensions. Inset: gold colloid before (left vial) and after (right vial) the addition to a suspension of phages exposing the Au-binding peptides (M13 Au-6).

**Figure 3 ijms-25-11222-f003:**
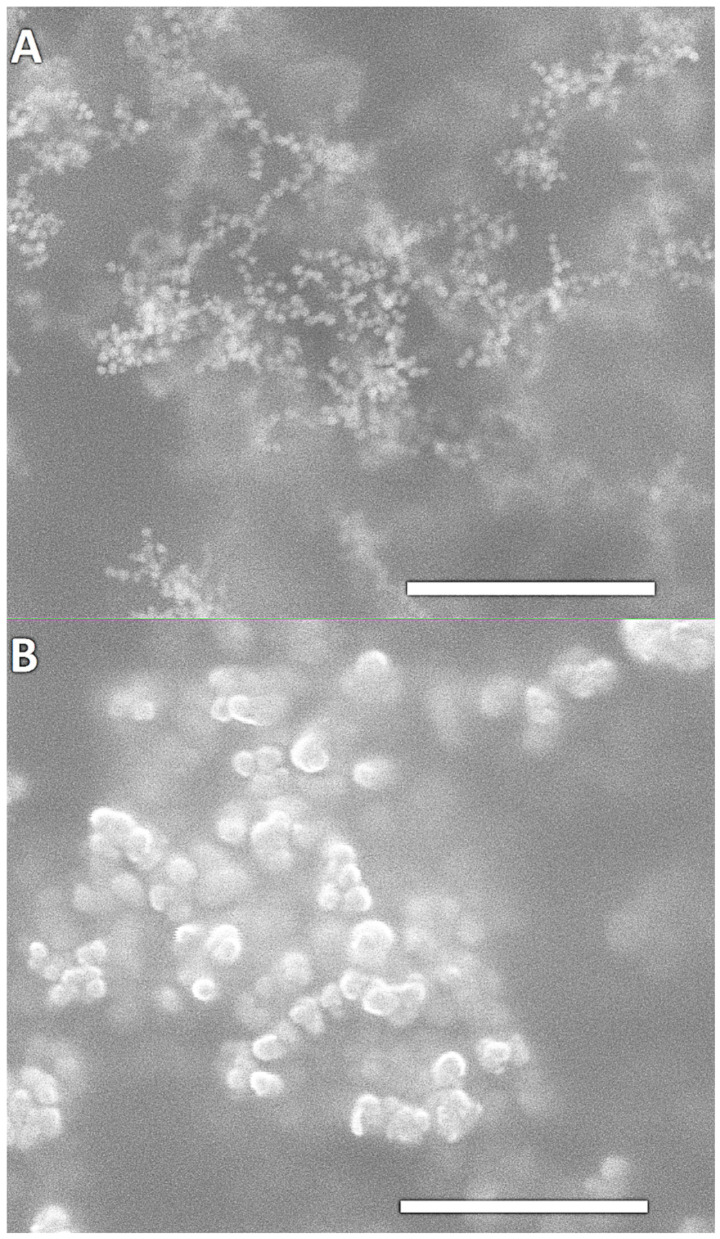
SEM images of Au nanoparticles (**A**) before and (**B**) after the addition to phages exposing the Au-binding peptides (M13 Au-6). Scale bars: 500 nm.

**Figure 4 ijms-25-11222-f004:**
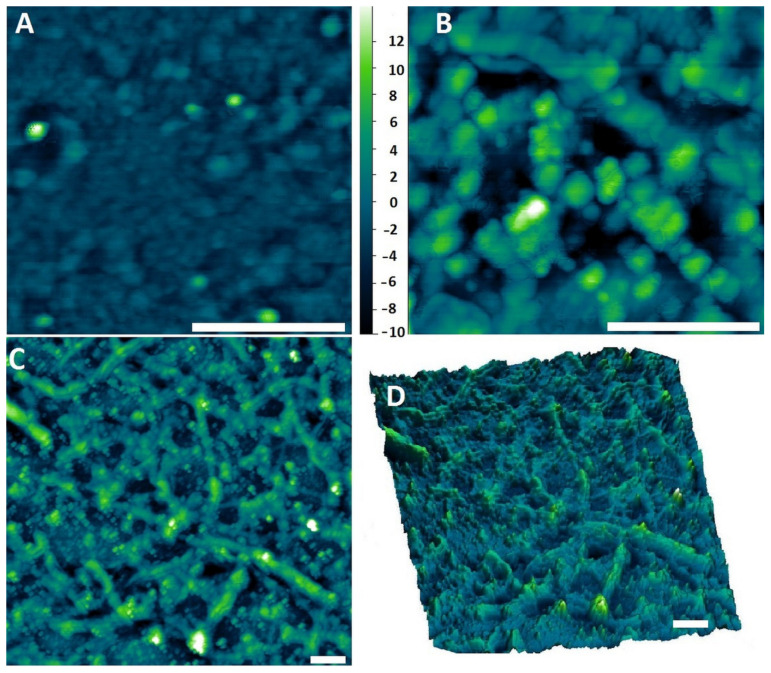
AFM images of (**A**) gold nanoparticles on a glass support, (**B**,**C**) M13 Au-6 phages on a glass support covered with AuNPs. (**D**) Three-dimensional AFM image of M13 Au-6 phages on a glass support covered with AuNPs. Scale bars: 500 nm.

**Figure 5 ijms-25-11222-f005:**
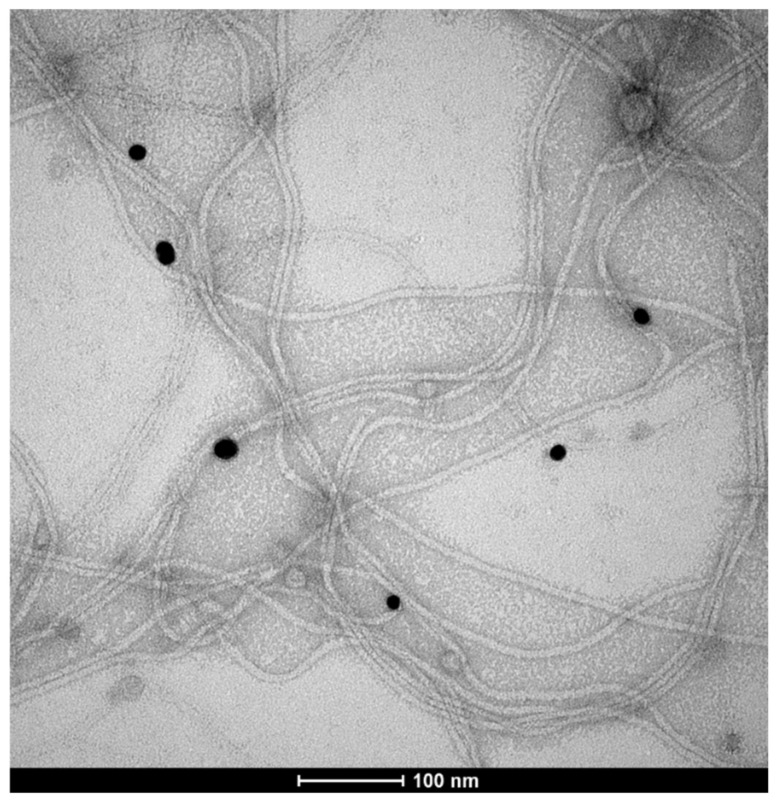
TEM image of M13 Au-6 phages exposing the Au-binding peptide on the pIII protein located at the virion end, interacting with preformed Au nanoparticles. The black spots represent AuNPs and the filamentous structures are M13 phages.

**Figure 6 ijms-25-11222-f006:**
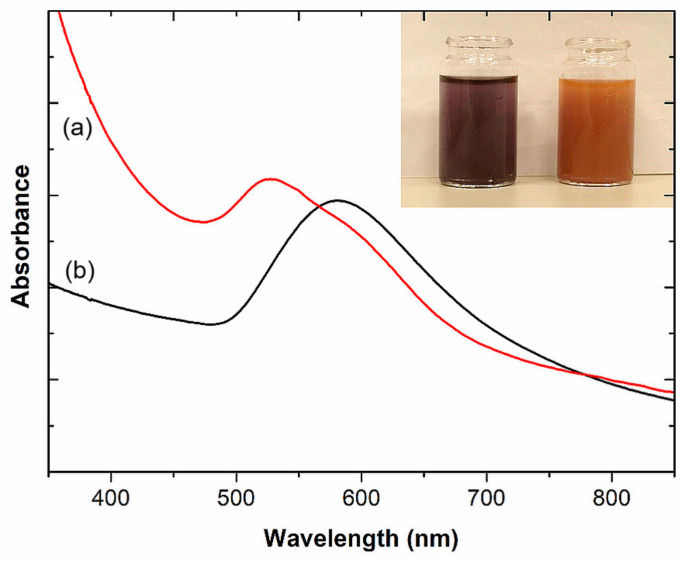
UV–vis spectra of gold nanoparticles synthesized by the M13 Au-6 phages in the presence of (a) ammonia and (b) triethylamine. Inset: vials with gold colloids synthesized in the presence of ammonia (left vial) or TEA (right vial), both used in all experiments for pH adjustment.

**Figure 7 ijms-25-11222-f007:**
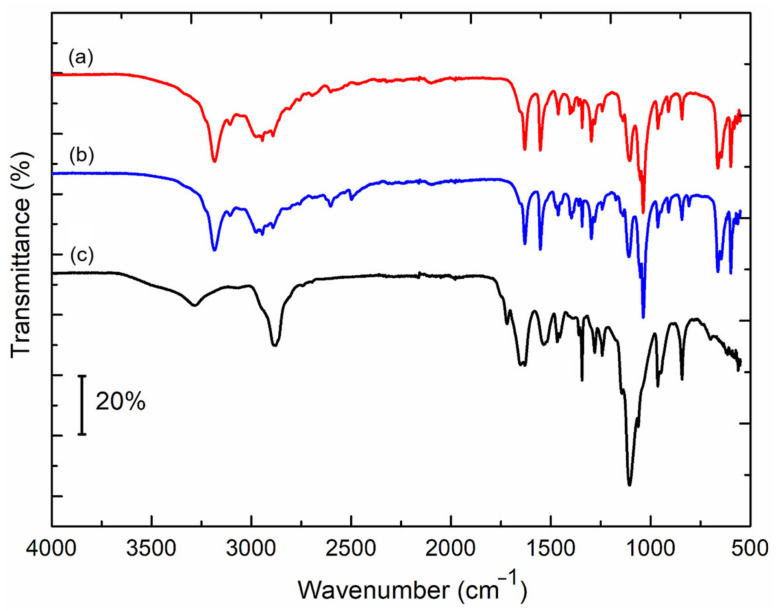
FTIR spectra of (a) M13 Au-6 phages with AuNPs synthesized in the presence of ammonia, (b) M13 Au-6 phages with AuNPs synthesized in the presence of triethylamine, and (c) M13 Au-6 phages before AuNP synthesis.

**Figure 8 ijms-25-11222-f008:**
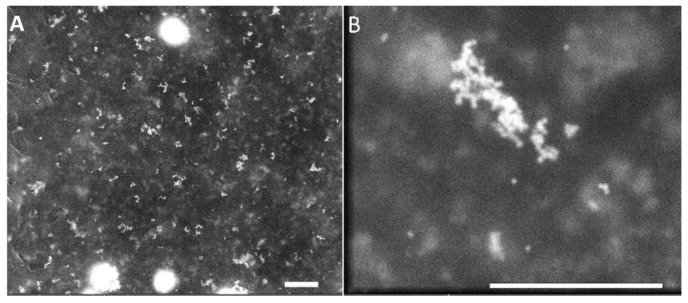
SEM images of Au nanoparticles synthesized by the M13 Au-6 phages in the presence of ammonia. Due to the nature of the material, which is mostly organic, there is a blurring (clouding) effect in the SEM images. To avoid destroying the phage–Au bioconjugates, no special manipulation (to eliminate this effect) was performed prior to SEM imaging. Magnifications: (**A**) 25,000× and (**B**) 150,000×. Scale bars: 500 nm.

**Figure 9 ijms-25-11222-f009:**
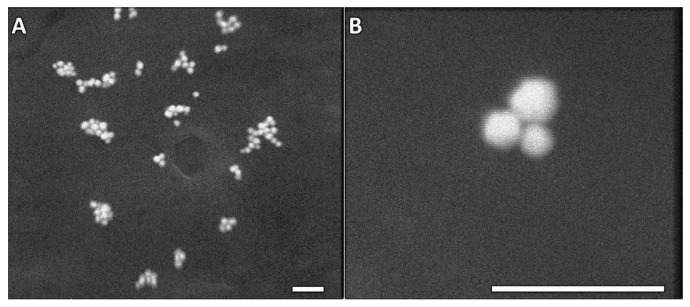
SEM images of Au nanoparticles synthesized by the M13 Au-6 phages in the presence of triethylamine. Magnifications: (**A**) 25,000× and (**B**) 150,000×. Scale bars: 500 nm.

**Figure 10 ijms-25-11222-f010:**
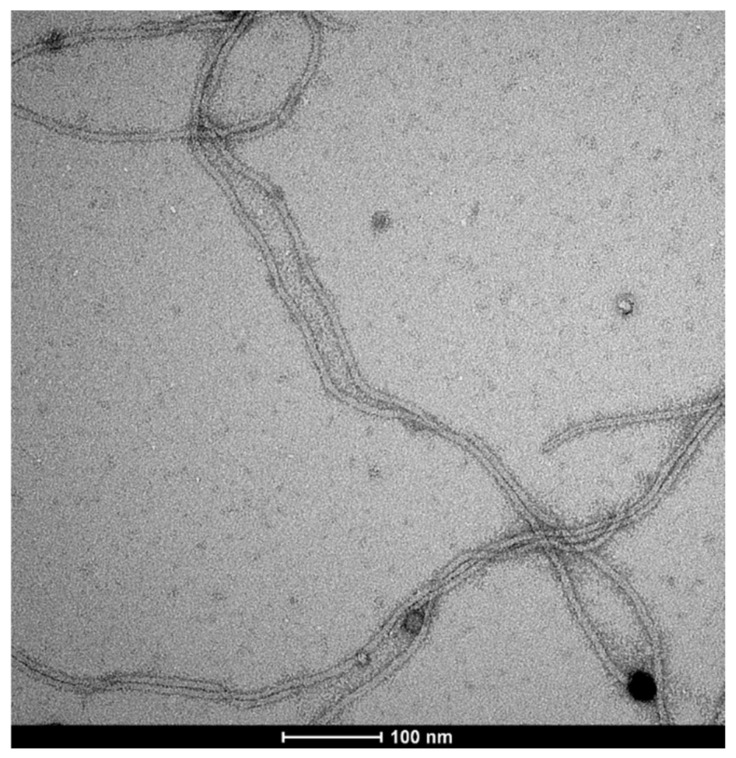
Exemplary TEM image of Au nanoparticles synthesized by the M13 Au-6 phages. The M13 Au-6 phages and an Au nanoparticle are shown as long filamentous structures and a black spot, respectively.

**Figure 11 ijms-25-11222-f011:**
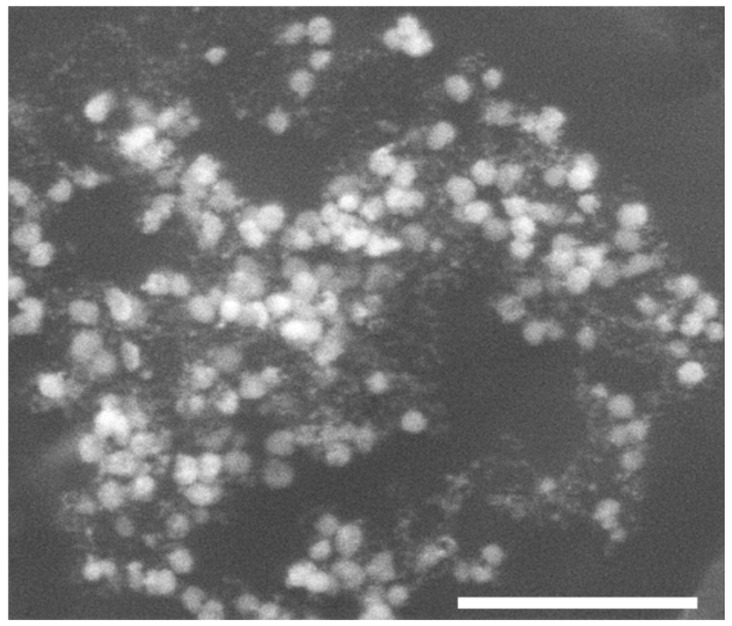
SEM image of Au nanoparticles synthesized, in the presence of TEA, by the chemically synthesized, gold-binding peptide (HLYLNTASTHLG) only. Scale bar: 1 µm.

**Table 1 ijms-25-11222-t001:** Identified peptide sequences exposed on the pIII protein of the M13 phage after three rounds of a biopanning procedure.

Clone No.	Sequence	pI ^α^	MW [Da]
M13 Au-1	PPLLKVSSHQLS	9.18	1305.54
M13 Au-2	LTMRISNEMAWV	6.00	1450.73
M13 Au-3	YAHWISPDQPPY	5.08	1473.61
M13 Au-4	HQWDYAVKYVGS	6.74	1452.59
M13 Au-5	FLPPVSRFPTPY	8.75	1420.67
M13 Au-6	HLYLNTASTHLG	6.92	1326.47
M13 Au-7	FNLPQGFDAIPS	3.80	1305.45
M13 Au-8	NTADAWSGRITA	5.84	1262.34
M13 Au-9	HLYLNTASTHLG	6.92	1326.47
M13 Au-10	GTRPSVQVLANV	9.75	1240.43
M13 Au-11	GHSVREAFDEYG	4.65	1366.41
M13 Au-12	KYLRPTWLVDDH	6.75	1542.76

^α^ Isoelectric point (pI) and molecular weight (MW) values were calculated using the Compute pI/MW tool available online https://www.expasy.org/resources/compute-pi-mw (accessed on 1 September 2024).

## Data Availability

All major data are included in this paper. Raw data are available from the authors upon request.

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
