# Peer review of "Engineered M13-Derived Bacteriophages Capable of Gold Nanoparticle Synthesis and Nanogold Manipulations"

_ijms, 2024, doi:10.3390/ijms252011222_

Round 1

Reviewer 1 Report

Comments and Suggestions for Authors

This manuscript reported the development of a M13 phage with Au-binding peptides via the phage display method, which can be used to synthesize Au under ambient condition. This M13 Au-6 phage has the specific peptide sequence (HLYLNTASTHLG) that can bind to Au surface due to the rich -OH group. However, there are several major issues that make this manuscript unsuitable to be published in Int. J. Mol. Sci. in its current form:

1.      The biggest concern is the novelty and significance of this work, since the synthesis of Au nanoparticles using biomaterials via green chemistry has been extensively documented in many literatures (e.g.: "Gold nanoparticles: Biosynthesis and potential of biomedical application." Journal of Functional Biomaterials 12, no. 4 (2021): 70. "On recent developments in biosynthesis and application of Au and Ag nanoparticles from biological systems." Journal of Nanotechnology 2022, no. 1 (2022): 5560244.).

2.      The discussion section is overly lengthy and includes a considerable amount of information that is not directly relevant to the presented results. This style of writing is more appropriate for a review article than for the discussion section of a research paper.

3.      The authors mention several potential applications of phage-assisted Au nanoparticle aggregation, including aggregation-induced emission enhancement, SERS enhancement, and enhanced photothermal therapy. However, it remains questionable whether these properties can be achieved within their system. The manuscript would be improved if the authors could demonstrate one or more of these properties experimentally.

4.      It is difficult to determine from the images in Figures 5 and 10 whether Au nanoparticles are attached to the tip of the phage virion. Authors themselves also acknowledged that “However, based on the TEM images it is difficult to unambiguously confirm that direct interactions between the pIII protein and AuNPs occur” (line 263). Therefore, the statement on line 181, “It is easy to see that AuNPs are located at the end of the phage virion structures,” is not appropriate and should be revised or removed.

Author Response

COMMENT 1:      The biggest concern is the novelty and significance of this work, since the synthesis of Au nanoparticles using biomaterials via green chemistry has been extensively documented in many literatures (e.g.: "Gold nanoparticles: Biosynthesis and potential of biomedical application." Journal of Functional Biomaterials 12, no. 4 (2021): 70. "On recent developments in biosynthesis and application of Au and Ag nanoparticles from biological systems." Journal of Nanotechnology 2022, no. 1 (2022): 5560244.).

RESPONSE 1: While we appreciate the Reviewer’s feedback, we respectfully disagree. We think this study makes a valuable contribution to the field as this is the first report of AuNPs obtained at neutral pH and room temperature using an engineered M13 phage system that simultaneously (i) relies on a novel Au-binding peptide experimentally selected in vitro for its high gold-binding affinity (as opposed to using an arbitrarily chosen, potentially binding amino acid sequence); (ii) employs the engineered phage itself as the reducing agent in the preparation of AuNPs from HAuCl4; and (iii) enables biogenic synthesis of the Au-binding peptide by propagating the engineered phage in a bacterial culture, which is an easily scalable and environmentally benign method. This explanation has been added to the revised manuscript (lines 333-340).

To further emphasize the limitations of previous reports and the novelty of our work, the following paragraph has also been added:

“Such biomolecule-based approaches have been increasingly used due to stringent environmental regulations and to maintain the biocompatibility of AuNPs for downstream applications [17]. However, although these methods employ biomolecules as one of the reagents in AuNP synthesis, in most cases they also require large amounts of hazardous, highly toxic chemicals; strong, conventional reducing agents such as NaBH4; or harsh conditions, including extreme pH [21]. For example, peptides that have been commonly used as green alternatives for AuNP preparation have in fact been routinely obtained by chemical synthesis methods such as Boc and Fmoc solid-phase syntheses, which involve the use of large volumes of hazardous solvents, rendering the "greenness" of the subsequent peptide-directed synthesis of AuNPs questionable [22,23]. Therefore, there is still room for improvement in such biogenic methods of AuNPs synthesis.” (lines 70-80).

In addition, the distinctive features of the gold-binding peptides exposed on the phage surface in our study were indicated in the last paragraph of the original Discussion (lines 462-475), which after revision reads as follows:

“This phage-peptide system combined the ability of the pIII phage coat protein to self-assemble into a well-defined architecture with the precise gold recognition feature of the HLYLNTASTHLG peptide. Unlike previous reports on the fabrication of gold nanoparticles using phage-display-selected peptides, this study features the gold-binding phage that has a dual role in the syntheses. First, propagation of the peptide-presenting phage enables biogenic synthesis of the desired peptide. Second, the propagated, peptide-presenting phage then enables the fabrication and manipulation of gold nanoparticles. In contrast, previous works on the fabrication of gold nanostructures reported that the selected peptide was chemically synthesized after the phage display selection step, or the peptide-presenting phage was used only as a scaffold for the assembly of preformed AuNPs into other nanostructures, and not as a reducing agent in the synthesis of AuNPs from the HAuCl4 precursor”.

Furthermore, we have added the suggested references to the Introduction section, although neither of the two reviews describes the use of viruses or peptides for the green synthesis of nanoparticles.

COMMENT 2:      The discussion section is overly lengthy and includes a considerable amount of information that is not directly relevant to the presented results. This style of writing is more appropriate for a review article than for the discussion section of a research paper.

RESPONSE 2: We agree that the original manuscript contained quite a lengthy discussion. Perhaps we wanted to discuss the subject too deeply which, however, led to lower clarity of the text. According to the Reviewer’s recommendation, Discussion has been substantially shortened (by 88 lines, corresponding to over 1000 words), and in the revised manuscript it is focused on aspects strictly connected to the results presented in our report.

COMMENT 3:      The authors mention several potential applications of phage-assisted Au nanoparticle aggregation, including aggregation-induced emission enhancement, SERS enhancement, and enhanced photothermal therapy. However, it remains questionable whether these properties can be achieved within their system. The manuscript would be improved if the authors could demonstrate one or more of these properties experimentally.

RESPONSE 3: We appreciate this comment, and we fully agree that a demonstration of the applicability of the phage-assisted Au nanoparticle aggregation would be of great interest. However, one must consider that such a demonstration would need a huge amount of work and a relatively long time, and in fact, it would be a separate story, suitable for an independent research report (another article). Analyzing the literature, on could find that it is a common approach to publish the properties of newly developed materials first, which is then followed by subsequent works on their specific applications. Nevertheless, we also agree that the use of the gold-binding phages described in our manuscript is hypothetical at this stage of the research. Therefore, in the revised manuscript we indicated this clearly, for example, by modifying the sentences, like “The phage-peptide system could synthesize AuNPs, aggregate them, as well as precipitate gold from a colloid and appears to be potentially useful for various manipulations of gold nanostructures” (lines 464-466), stressing the potential, and not the actual, use of the materials.

COMMENT 4:      It is difficult to determine from the images in Figures 5 and 10 whether Au nanoparticles are attached to the tip of the phage virion. Authors themselves also acknowledged that “However, based on the TEM images it is difficult to unambiguously confirm that direct interactions between the pIII protein and AuNPs occur” (line 263). Therefore, the statement on line 181, “It is easy to see that AuNPs are located at the end of the phage virion structures,” is not appropriate and should be revised or removed.

RESPONSE 4: We agree that the previous statement (line 181 of the original manuscript) was too strong, and could be an overinterpretation of the results. Therefore, we have modified the text which in the revised manuscript reads as follows (lines 226-229): “Many AuNPs colocalized with the terminal parts (ends) of the filamentous phage virion structures. This observation confirmed that it is likely that M13 Au-6 phage clones interact with AuNPs through a virion tip where the modified pIII protein is located.”

Reviewer 2 Report

Comments and Suggestions for Authors

The author reported the Engineered M13-derived bacteriophages capable of gold nanoparticle synthesis and nanogold manipulations. The manuscript presents good results but some improvement is highly required specifically highlighting the aim of the work and key finding..

1.      In the abstract add results and highlight the key implications of this study, show some results with number’s, Abstract is very generally written

2.      In introduction clearly mention some limitation in previous study and explain how this study overcomes those limitation with novelty of using Engineered M13-derived bacteriophages

3.      Include concentration and time of gold syntesisi. Provide all the details

4.      Compare the binding efficiency of the engineered bacteriophages to other gold-binding peptides and explain how this research is useful

5.      Explain the reason behind observed red shift in the UV-Vis spectra and its relation to the nanoparticle size and aggregation in details.

6.      Some figures are not clear. Improve the clarity of the figures, particularly the SEM and TEM images,

7.      Author mentioned in Line 151 for SEM Note that both sample images are magnified. Can you explain how you magnified? Images are not clear and clouds are seen , I suggest author should present the original images without magnification to see clear

8.      Recheck for grammatical errors

Comments on the Quality of English Language

Minor check spelling are required

Author Response

COMMENT 1:      In the abstract add results and highlight the key implications of this study, show some results with number’s, Abstract is very generally written

RESPONSE 1: We have added the suggested content to the Abstract, which now reads as follows (lines 16-30):  “For years, gold nanoparticles (AuNPs) have been widely used in medicine and industry, creating the need for effective methods for their preparation and manipulation. Although various experimental procedures have been reported to address these needs, none of them is optimal for all purposes. In this work, we engineered the N-terminus of the pIII minor coat protein of bacteriophage (phage) M13 to expose a novel HLYLNTASTHLG peptide that effectively and specifically binds gold. In addition to binding gold, these engineered phages could synthesize spherical AuNPs of 20 nm and other sizes depending on the reaction conditions, aggregate them, and precipitate gold from a colloid, as revealed by transmission electron microscopy (TEM), atomic force microscopy (AFM), and scanning electron microscopy (SEM), as well as ultraviolet-visible (UV-vis) and Fourier-transform infrared (FTIR) spectroscopic methods. We demonstrated that engineered phages exposing foreign peptides selected from a phage-displayed library may serve as sustainable molecular factories for both the synthesis of the peptides and the subsequent bioreduction of gold ions at room temperature and neutral pH in the absence of strong reducing agents, such as commonly used NaBH4. Collectively, the results suggest the potential applicability of the engineered phages and the new, in vitro-identified gold-binding peptide in diverse biomimetic AuNP manipulations”.

COMMENT 2:      In introduction clearly mention some limitation in previous study and explain how this study overcomes those limitation with novelty of using Engineered M13-derived bacteriophages

RESPONSE 2: We have added the following suggested content to the Introduction (lines 70-80): “Such biomolecule-based approaches have been increasingly used due to stringent environmental regulations and to maintain the biocompatibility of AuNPs for downstream applications [17]. However, although these methods employ biomolecules as one of the reagents in AuNP synthesis, in most cases they also require large amounts of hazardous, highly toxic chemicals; strong, conventional reducing agents such as NaBH4; or harsh conditions, including extreme pH [21]. For example, peptides that have been commonly used as green alternatives for AuNP preparation have in fact been routinely obtained by chemical synthesis methods such as Boc and Fmoc solid-phase syntheses, which involve the use of large volumes of hazardous solvents, rendering the "greenness" of the subsequent peptide-directed synthesis of AuNPs questionable [22,23]. Therefore, there is still room for improvement in such biogenic methods of AuNPs synthesis.”

This new content works well with the existing one (lines 112-116): “In this work, with the use of phage display, we aimed to develop an engineered M13 phage that exposes a new, in vitro-identified Au-binding peptide capable of synthesizing AuNPs under benign conditions. Such phages could be employed for biogenic, bottom-up synthesis and manipulation of gold nanostructures particularly for further biological applications, which have considerable concerns over contamination”.

COMMENT 3:      Include concentration and time of gold syntesisi. Provide all the details

RESPONSE 3: We are grateful to the reviewer for pointing this out. Indeed, the conditions of gold synthesis should be provided in more detail.

In this work, nanogold was synthesized using two methods. First, to determine the gold binding efficiency of the engineered phages, gold nanoparticles were obtained from HAuCl4 by a well-established chemical reduction with citrate (the Turkevich method). After confirming the efficient gold binding by the M13 Au-6 phage, the phage inherent ability to synthesize AuNPs from HauCl4 in the absence of conventional chemical reductants, such as citrate or NaBH4, was explored and proved successful.

While a detailed description of the phage-assisted synthesis process was included in the Materials and Methods section of the original manuscript and the information provided is sufficient to fully reproduce the experiment, we agree that the conditions of the process should also be included in the main text. Therefore, such a description, including reaction time and pH, has been added to both the Results and the Discussion sections (lines 237-240, and 418-420, respectively). The text in Results now reads as follows: “Au nanoparticles were obtained overnight from the HAuCl4 precursor in the reaction with engineered phages (M13 Au-6, 1011 pfu/mL) both in the presence of ammonia and triethylamine (TEA) which were used to adjust pH to neutral”. The text in Discussion now reads as follows: “Engineered phages (1011 pfu/mL) formed AuNPs from a gold ion precursor overnight at RT and neutral pH in the presence of ammonia or TEA and in the absence of strong reducing agents, such as NaBH4”.

The chemical reduction method used to prepare AuNPs for binding efficiency evaluation was described in the “A chemical method for gold colloid synthesis” subsection of Materials and Methods. This part has also been improved and now reads as follows (lines 505-509): “Briefly, 30 mg of HAuCl4·xH2O (49% wt. Au) was dissolved in 150 mL of deionized water and heated under reflux for 5 min. Then, 1.8 mL of trisodium citrate solution (50 mg/mL) was added with vigorous stirring. Heating was continued until a stable purple color was obtained (c.a. 2 h)”.

COMMENT 4:      Compare the binding efficiency of the engineered bacteriophages to other gold-binding peptides and explain how this research is useful

RESPONSE 4:  We added the following fragment to the Results section (lines 138-145):
“In addition, by comparing the output/input (O/I) ratios, we found that the phage-presented dodecapeptide identified in this study had a higher affinity for gold than a previously reported, phage-display-selected TLLVIRGLPGAC dodecapeptide, which had been characterized as having a strong gold-binding affinity [38]. With an O/I ratio of approximately 7 × 10-3, the phage-presented HLYLNTASTHLG peptide selected in our study, exhibited approximately seven times higher gold binding efficiency than its TLLVIRGLPGAC counterpart, which had an O/I ratio of approximately 1 × 10-3.”  In this light, we concluded that the method and the obtained recombinant bacteriophage producing the HLYLNTASTHLG can be useful in any processes and technologies requiring  gold binding, as being more efficient in this aspect than previously reported peptides. This is also indicated in the revised manuscript (lines 410-417).

COMMENT 5:      Explain the reason behind observed red shift in the UV-Vis spectra and its relation to the nanoparticle size and aggregation in details.

RESPONSE 5: We thank the reviewer for this suggestion. We have modified the Results section to fully explain this phenomenon (lines 156-182):

“After the addition of the gold colloid to the phage suspension, aggregation of the nanoparticles occurred, which was directly observed as a color change from deep red to bluish (see Figure 2, inset), and confirmed by absorption spectroscopy measurements (Figure 2). Gold nanoparticles exhibit a distinctive optical feature commonly referred to as localized surface plasmon resonance (LSPR), which is a result of the collective oscillation of conduction electrons on the metal surface when excited by light. In the case of spherical AuNPs, their optical properties are highly dependent on both the diameter of the nanoparticles themselves and the physical properties of their surroundings. Smaller gold nanospheres primarily absorb light, with absorption bands around 520 nm (as can be seen in Figure 2, curve a). When AuNPs form agglomerates of higher dimensions than individual nanoparticles, they exhibit enhanced scattering, with bands broadening and shifting towards longer wavelengths (a phenomenon known as red-shifting, Figure 2, curve b). This enhanced scattering in agglomerated particles is due to both their larger optical cross-sections and the increase in the ratio of scattering to total extinction with increasing particle size. As mentioned above, the optical properties of gold nanoparticles are also affected by the refractive index of their surrounding environment. When the refractive index near the nanoparticle surface increases, the absorption band of the nanoparticle also shifts towards longer wavelengths (a red-shift). The red-shift of c.a. 30 nm, observed in the UV-Vis spectrum of a mixture of AuNPs and phages relative to the spectrum of AuNPs alone, can be attributed to the enhanced scattering, observed for agglomerates, and to the increase in the refractive index near the AuNP surface resulting from the interaction between the phages and gold. Such an observation reflects the effect of nanoparticle size and local refractive index on the optical properties of AuNPs which is widely applied in the colorimetric detection of biological molecular interactions that alter the aggregation and dispersion status of particles [39]. For comparison, the UV-vis spectrum of the phage suspension is also shown in Figure 2 (curve c), revealing no distinct absorption band in the analyzed region”.

COMMENT 6:      Some figures are not clear. Improve the clarity of the figures, particularly the SEM and TEM images,

RESPONSE 6: Due to the nature of the material, which is mostly organic, it was difficult (if not impossible) to obtain clear SEM images. Since we did not want to destroy the phage-Au bioconjugates, no special manipulation was performed prior to SEM imaging. We believe that the SEM images are still informative as AuNPs are clearly visible in the images. The blurring is due to the presence of phages/peptides surrounding the AuNPs (see lines 196-201, 291-293).

While we agree with the Reviewer that blurring/clouds are visible in the SEM images, we consider the TEM images to be of sufficiently good quality to be demonstrated in the paper.

COMMENT 7:      Author mentioned in Line 151 for SEM Note that both sample images are magnified. Can you explain how you magnified? Images are not clear and clouds are seen , I suggest author should present the original images without magnification to see clear

RESPONSE 7: Thank you for bringing this to our attention. This comment alerted us to the potentially misleading original description of the imaging result. Both are original micrographs obtained with a scanning electron microscope at the same magnification. We apologize for the ambiguous original wording. The original sentence was intended to emphasize that the presented differences in the size of the AuNPs before and after the addition of the gold colloid to the phage suspension are not an effect of using different magnifications in SEM imaging. As explained in Response 6, the blurring effect can be attributed to the presence of phages/peptides surrounding the AuNPs.

The original sentence has been revised to read as follows (lines 189-191): “Both SEM images were captured at the same magnification to clearly show the size differences of AuNPs before and after the addition of the preformed gold colloid to the engineered phage suspension”.

COMMENT 8:      Recheck for grammatical errors

RESPONSE 8: As requested, the text of the paper has been polished by a professional English language editor.

Round 2

Reviewer 1 Report

Comments and Suggestions for Authors

The authors’ claim that “the engineered phage itself serves as the reducing agent in the preparation of AuNPs from HAuCl₄” (line 337) is not fully supported by the experimental evidence. In the control experiment for phage-assisted synthesis of AuNPs, the formation of metallic Au without the phages exposing the Au-binding peptides indicates that the Au(III) ions can be reduced in the presence of ammonium/triethylamine only.  This raises uncertainty about whether the Au-binding peptides function as both the reducing agent and capping ligand, or only as the capping ligand. The authors’ statement on line 337 would be more convincing if they could demonstrate that Au(III) ions are reduced to Au in the presence of the Au-binding peptides alone, without ammonium or triethylamine.

Author Response

COMMENT 1: The authors’ claim that “the engineered phage itself serves as the reducing agent in the preparation of AuNPs from HAuCl₄” (line 337) is not fully supported by the experimental evidence. In the control experiment for phage-assisted synthesis of AuNPs, the formation of metallic Au without the phages exposing the Au-binding peptides indicates that the Au(III) ions can be reduced in the presence of ammonium/triethylamine only.  This raises uncertainty about whether the Au-binding peptides function as both the reducing agent and capping ligand, or only as the capping ligand. The authors’ statement on line 337 would be more convincing if they could demonstrate that Au(III) ions are reduced to Au in the presence of the Au-binding peptides alone, without ammonium or triethylamine.

RESPONSE 1: We agree that such experiments seem optimal to demonstrate the reducing role of the peptide-presenting phage directly. However, to ensure virus-friendly conditions and to attempt phage-mediated synthesis from the HAuCl4 precursor at all, one needs to raise the pH of the aqueous chloroauric acid solution with one or another alkali. We performed the phage-mediated syntheses after using two different bases. As explained in the Materials and Methods section, drops of these bases were added to the HAuCl4 • aq to adjust the pH to neutral. The volume of ammonia/TEA added dropwise to neutralize the pH is small relative to the total reaction volume. Therefore, in our opinion, it is rather unlikely that either TEA or ammonia played a key role in the overnight reduction of gold ions to AuNPs, although it cannot be excluded that they could be somehow implicated in such reduction. Previous studies on the role of TEA in the synthesis of sucralose-capped Ag nanoparticles or Ag and Ag/Au bimetallic nanoparticles in glycerol reported that no nanoparticles were formed from the noble metal ion precursors at a low concentration of TEA (Filippo et al.. Controlled synthesis and chain-like self-assembly of silver nanoparticles through tertiary amine. Colloids and Surfaces A: Physicochem. Eng. Aspects. 2013, 417, 10-17; and Nalawade et al. Triethylamine induced synthesis of silver and bimetallic (Ag/Au) nanoparticles in glycerol and their antibacterial study. J Nanostruct Chem. 2014, 4:113, respectively).

Importantly, in our report, Au nanoparticles were obtained from the HAuCl4 precursor in the reaction with engineered phages (M13 Au-6) overnight. In a previously published work, which reported the use of TEA as the sole reducing agent to synthesize AuNPs from HAuCl4, a large number of nanoparticles was obtained from the combined aqueous solution of HAuCl4 and TEA after 20+ days  (Kuo and Chen. Generation of Gold Thread from Au(III) and Triethylamine. Langmuir. 2006, 22, 7902-7906).

Furthermore, as mentioned in the Discussion of the original manuscript, all amino acids except cysteine can reduce gold ions to nanoparticles. We have added more references in the revised manuscript to emphasize that this ability of amino acids was proved in numerous previous papers (line 429 of the revised manuscript).

In summary, combining the results of these previous reports with the results of our work we believe that – in addition to its capping role - the peptide-presenting phage did act as a reducing agent in the overnight synthesis of AuNPs, even if TEA or ammonia also played some role in the reduction of gold ions to AuNPs.

The sentence quoted by the Reviewer has been modified to read: “The phage system (…) utilizes the properties of the engineered phage itself (as opposed to the use of a strong chemical reductant) in the preparation of AuNPs from HAuCl4” (lines 337-339 of the revised manuscript).

We also clarified throughout the manuscript that i) TEA/ammonia was added dropwise to adjust the pH to neutral; ii) under the conditions of our study, the nanoparticles were formed from HAuCl4 overnight; and iii) we demonstrate the possibility of obtaining nanoparticles without the need to employ strong reducing agents such as NaBH4 that is routinely used for the facile synthesis of AuNPs from HAuCl4.

Reviewer 2 Report

Comments and Suggestions for Authors

Article can be accepted now

Author Response

COMMENT 1: Article can be accepted now

RESPONSE 1: Thank you for taking the time to review our manuscript and the insightful comments you made throughout the reviewing process.